# *Cis*-regulatory differences in isoform expression associate with life history strategy variation in Atlantic salmon

Jukka-Pekka Verta [1,2]*, Paul Vincent Debes [1,2¤], Nikolai Piavchenko[1], Annukka Ruokolainen[1], Outi Ovaskainen [1], Jacqueline Emmanuel Moustakas-Verho [1,2], Seija Tillanen[1,2], Noora Parre[1], Tutku Aykanat [1,2], Jaakko Erkinaro [3], Craig Robert Primmer [1,2]*

**1** Organismal and Evolutionary Biology Research Programme, University of Helsinki, Helsinki, Finland, **2** Institute of Biotechnology, University of Helsinki, Finland, **3** Natural Resources Institute Finland (LUKE), Oulu, Finland

¤ Current address: Department of Aquaculture and Fish Biology, Hólar University College, Iceland
* jukka-pekka.verta@helsinki.fi (JPV); craig.primmer@helsinki.fi (CRP)

**Data Availability Statement:** Data and code are available in Dryad repository (10.5061/dryad. k6djh9w4w).

## Abstract

A major goal in biology is to understand how evolution shapes variation in individual life histories. Genome-wide association studies have been successful in uncovering genome regions linked with traits underlying life history variation in a range of species. However, lack of functional studies of the discovered genotype-phenotype associations severely restrains our understanding how alternative life history traits evolved and are mediated at the molecular level. Here, we report a *cis*-regulatory mechanism whereby expression of alternative isoforms of the transcription co-factor *vestigial-like 3* (*vgll3*) associate with variation in a key life history trait, age at maturity, in Atlantic salmon (*Salmo salar*). Using a common-garden experiment, we first show that *vgll3* genotype associates with puberty timing in one-year-old salmon males. By way of temporal sampling of *vgll3* expression in ten tissues across the first year of salmon development, we identify a pubertal transition in *vgll3* expression where maturation coincided with a 66% reduction in testicular *vgll3* expression. The *late* maturation allele was not only associated with a tendency to delay puberty, but also with expression of a rare transcript isoform of *vgll3* pre-puberty. By comparing absolute *vgll3* mRNA copies in heterozygotes we show that the expression difference between the *early* and *late* maturity alleles is largely *cis*-regulatory. We propose a model whereby expression of a rare isoform from the *late* allele shifts the liability of its carriers towards delaying puberty. These results exemplify the potential importance of regulatory differences as a mechanism for the evolution of life history traits.

## Author summary

Alternative life history strategies are an important source of diversity within populations and promote the maintenance of adaptive capacity and population resilience. However, in

**Funding:** Financial support was provided by the University of Helsinki, Academy of Finland (grant numbers 314254 and 314255) and the European Research Council under the European Union's Horizon 2020 research and innovation programme (grant agreement number 742312). The funders had no role in study design, data collection and analysis, decision to publish, or preparation of the manuscript.

**Competing interests:** The authors have declared that no competing interests exist.

many cases the molecular basis of different life history strategies remains elusive. Age at maturity is a key adaptive life history trait in Atlantic salmon and has a relatively simple genetic basis. Using salmon age at maturity as a model, we report a mechanism whereby different transcript isoforms of the key age at maturity gene, *vestigial-like 3* (*vgll3*), associate with variation in the timing of male puberty. Our results show how gene regulatory differences in conjunction with variation in gene transcript structure can encode for complex alternative life histories.

## Introduction

Some of the most ecologically and evolutionarily important traits involve alternative life-history strategies and the trade-offs therein [1]. One such example is maturation age, whereby early maturing individuals avoid the increased risk of mortality associated with later reproduction, but often with the trade-off of lower fecundity [2]. Genome-wide association studies have been successful in uncovering genome regions controlling for life-history trait variation such as survival and reproductive success in Soay sheep [3], early versus late age at maturity in Atlantic salmon [4,5], early versus late run timing in Chinook salmon and steelhead trout [6–8], age and size at flowering in monkeyflowers [9] and alternative reproductive strategies in ruffs [10]. However, despite this progress mapping the genetic basis of life history traits, moving beyond genome-level associations to an understanding of the molecular basis of such variation in life history traits and trade-offs has thus far remained elusive. This lack of functional insight into the discovered genotype-phenotype associations severely restrains our understanding of the molecular basis of evolution in life history traits [11].

In Atlantic salmon, age at maturity is a key adaptive life history trait that is evolving under a trade-off between survival and fecundity. Later-maturing, and therefore larger, individuals have higher reproductive fitness at spawning than earlier maturing (smaller) individuals [12,13], but also have a higher risk of mortality prior to first reproduction [2]. Males mature, on average, earlier than females and at smaller size, which is indicative of a potential sexual conflict for age at maturity in the species [14,15]. Genetic variation in a single genome region including the transcription co-factor gene *vestigial-like 3* (*vgll3*) was recently found to explain nearly 40% of variation in sea-age at maturity in both male and female salmon across 57 Northern-European wild populations [4], making salmon age-at maturity a genetically tractable model to study the molecular bases of life-history trait variation. Further, variation in *VGLL3* also associated with age at puberty in humans [16,17], indicating that functional mechanisms of puberty timing may be shared across vertebrates. However, in all cases the functional basis of the association between genetic variation in *vgll3* and age at maturity remains unknown.

The pattern of association observed in natural salmon populations can be used to generate hypotheses pertaining the functional mechanism that translates genetic variation to different maturation phenotypes. The strongest association between age at maturity and *vgll3* genotype maps to a single nucleotide polymorphism (SNP) in a non-coding region approximately 7000 basepairs 3' of *vgll3*, followed in statistical significance by mutations in *vgll3* coding sequence [4,5]. This suggests that genetic variation in linked *cis*-regulatory sequences (e.g. enhancers) may underlie the functional differences among salmon *vgll3* genotypes by changing their expression level or pattern. In other species, a similar pattern of linked non-coding changes is often observed in association studies for life history traits [6,7,10,18,19], suggesting that regulatory changes may be a common driver of evolution in life history strategies. However, non-

coding and coding hypotheses pertaining the control of maturation age are nuanced and not mutually exclusive; enhancer mutations can have effects unlinked to their genotype (they can act in *trans*) [20,21] and non-coding mutations can alter transcript structure, stability and translation via alternative transcription start sites and splicing [22–24].

To uncover the functional genetic basis of age at maturity variation in Atlantic salmon, we first validated the *vgll3* association with male maturation timing in different genetic backgrounds and in common garden settings by using 32 controlled-crossed families with known *vgll3* genotypes. Uncertainty remains regarding the tissues and developmental stages during which *vgll3* expression potentially influences the maturation phenotype. *Vgll3* expression and function has been linked to diverse processes such as adipocyte differentiation and myogenesis in mice [25,26], immune function, adiposity and puberty timing in humans [16,17] and cell fate commitment in salmon tissues including adipose and gonads [27,28], but genotype comparisons, *vgll3* transcript structure or *cis*-regulation have not been investigated. We determined *vgll3* temporal expression dynamics during the first year of salmon development up to male maturation by way of a time-series sampling of ten tissues from a total of 273 individuals, characterized *vgll3* transcript structure and further tested the importance of *cis*-regulatory mechanisms among *vgll3* genotypes by assaying *vgll3* allele-specific expression in heterozygotes. Our results reveal a potential functional mechanism that links allele-specific isoform expression differences in *vgll3* to variation in maturation timing and thus may mediate variation in life history strategies.

## Results

### *vgll3* genotype predicts male maturation probability

To uncover the functional genetic basis of age at maturity variation in Atlantic salmon, we first validated that the genotype at the *vgll3* locus on chromosome 25 (termed "*E*" for early sea-age at maturity and "*L*" for late sea-age at maturity after previous association study [4]) is associated with age at maturity in controlled conditions by rearing 656 individuals with known *vgll3* genotypes from 32 families in common garden conditions (S1 Fig). We assessed all males for maturation status non-lethally at the start of the breeding season (nine months post-hatching) and modeled maturation probabilities based on observed frequencies of mature versus non-mature fish as a function of *vgll3* genotype (see Methods). *Vgll3*\**EE* individuals showed a 10 times higher frequency of maturation (20%) compared to individuals with *vgll3*\**LL* genotypes (2%, S2 Fig). Correspondingly, maturation probability for *vgll3*\**EE* males (20.7%) was significantly higher compared to males with *vgll3*\**LL* genotypes (3%) (probit 95% credible interval: 0.59 to 3.07, $P = 0.006$, *vgll3*\**EE* $n = 93$ and *vgll3*\**LL* $n = 92$) (Fig 1A). Maturation probability for males heterozygous for *vgll3* (15.4%) was closer to *vgll3*\**EE* males compared to *vgll3*\**LL* males, and significantly different from *vgll3*\**LL* males (probit 95% credible interval: 0.32 to 3.88, $P = 0.018$, *vgll3*\**EL* $n = 199$), but not *vgll3*\**EE* males.

We additionally modeled maturation probability as a function of *vgll3* genotype and fish length, hypothesizing that early growth positively affects maturation probability [29]. Indeed, maturation probability positively associated with fish length before the start of the breeding season (eight months post-hatching) (probit 95% credible interval: 0.36 to 0.94, $P = 1\text{e-}4$, Fig 1B). Irrespective of size effect, the *vgll3*\**EE* genotype was still associated with a higher maturation probability compared to the *vgll3*\**LL* genotype (probit 95% credible interval: 0.54 to 2.84, $P = 0.005$), as well as *vgll3*\**EL* compared to *vgll3*\**LL* (probit 95% credible interval: 0.26 to 3.6, $P = 0.026$).

These results demonstrate that the *vgll3*\**E* allele associated with young sea-age at maturity in natural populations is associated with earlier maturation also in controlled conditions, and

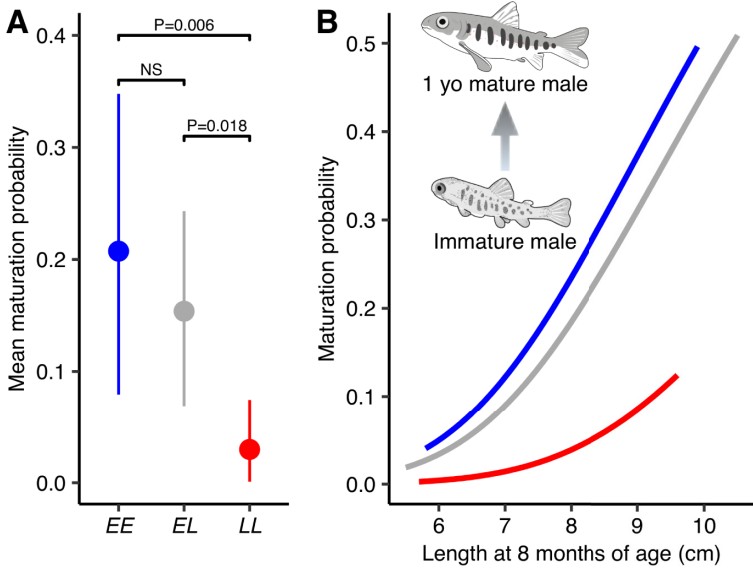

**Fig 1. *Vgll3* genotype associates with male maturation probability in controlled conditions.** (A) Probability of maturation at nine months post-hatching in controlled conditions is significantly higher in *vgll3*EE* individuals compared to *vgll3*LL* individuals. Points represent means and whiskers represent 95% credible intervals from a Bayesian Markov Chain Monte-Carlo animal model (see **Methods**). $N_{EE}$ = 98, $N_{EL}$ = 204, $N_{LL}$ = 82 (B) Maturation probability modeled as a function of fish length one month before the breeding season. Blue, *vgll3*EE*; grey, *vgll3*EL*; red, *vgll3*LL*.

also supports stronger dominance of *vgll3*E* allele as observed in males from natural populations [4]. Seeing that concurring results were observed in another wild-derived stock of Atlantic salmon [30], the genotype-phenotype association between *vgll3* and maturation timing is reproducible in controlled conditions and likely common in North-European strains of Atlantic salmon.

## *vgll3* expression correlates with maturation inhibition

Until now, it has remained unknown by which functional means, tissues, cell types and developmental time-points genetic variation in *vgll3* influences the maturation phenotype. In order to investigate the functional mechanisms resulting in earlier maturation in males carrying the *vgll3*EE* genotype and to establish the tissues and temporal time-points of relevance for *vgll3* expression, we first characterized *vgll3* transcript expression profiles of ten tissues and whole salmon juveniles in a time-series of samples including 13 time points and all *vgll3* genotypes by using reverse-transcription droplet digital PCR (RT-ddPCR) and a genotype-specific assay on exon 2 of the *vgll3* transcript (total *n* = 291, S3 Fig).

*Vgll3* expression in the heart was high relative to other tissues (Fig 2A), but also very variable, similar to results from three-year old juveniles from natural populations [28]. Expression in liver showed a linear downward trend from younger to older individuals, although the overall expression level remained very low relative to other tissues (S4 Fig). *Vgll3* expression in muscle was lowest among tissues, while immature ovaries showed intermediate and relatively stable expression (Fig 2A). Interestingly, while *vgll3* expression was high in immature testes, expression was reduced by 66% in mature testes compared to immature testes (37 copies/ng versus 107 copies/ng, i.e. 70 copies/ng fewer, 95% CI [47–93 fewer], two-tailed *t*-test, df = 25.9, *P* = 1e-06, $n_{mature}$ = 14, $n_{immature}$ = 65, Fig 2B). *Vgll3* expression was similarly lower in stripped

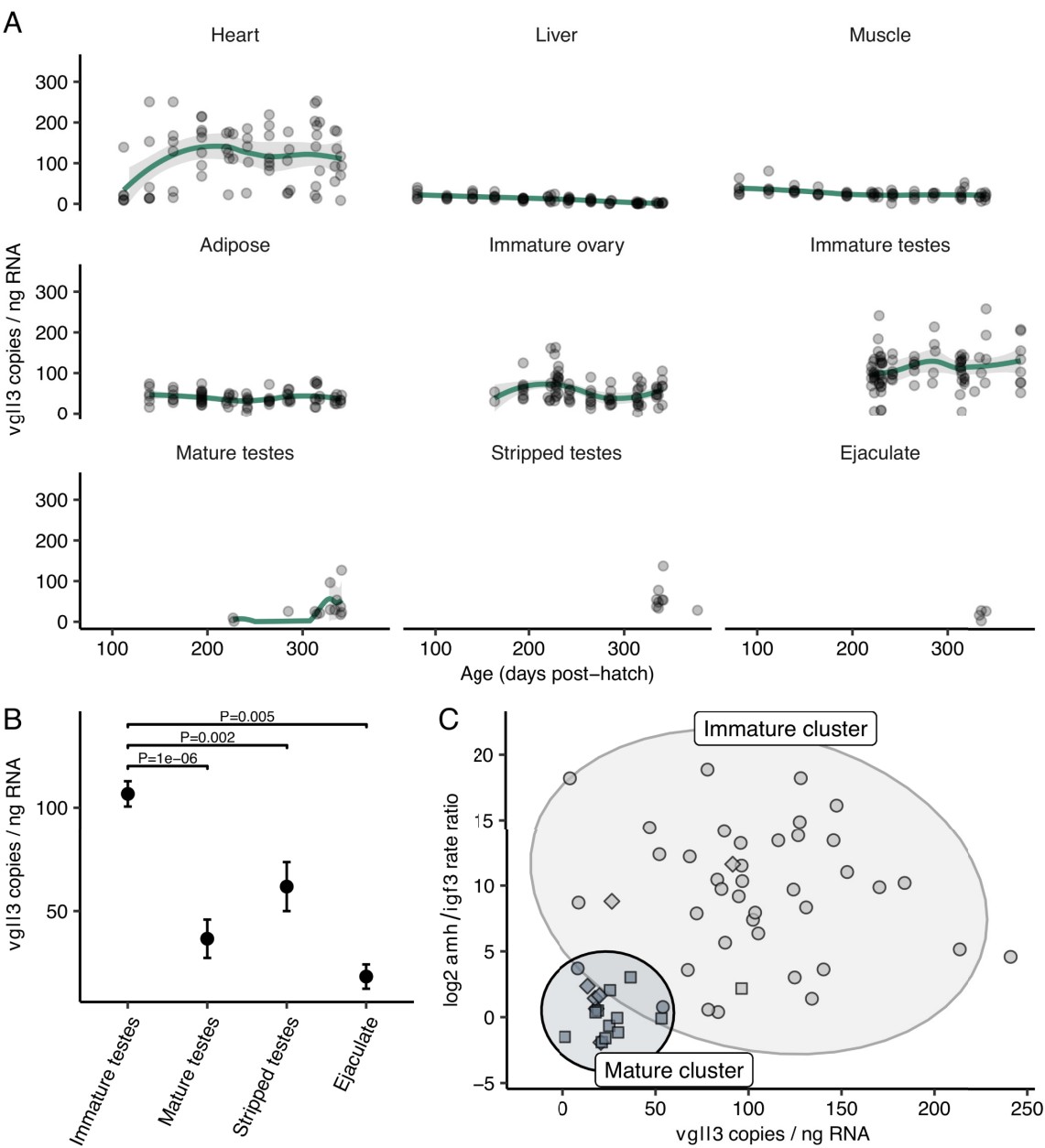

**Fig 2. *Vgll3* expression across salmon tissues during the first year of development.** (A) *Vgll3* reaches highest expression levels in the heart and immature testes. For sample size see S3 Fig. Line, loess smoothing; shading, 95% confidence interval. (B) Male maturation is associated with a 66% reduction in *vgll3* expression compared to immature testes expression levels. Reduction in *vgll3* expression is observed in all major tissues of the mature testes. Stripped testes; mature testes with ejaculate removed manually. Error bars show SE. (C) *Vgll3* expression shows a strong association with relative expression levels of *amh/igf3*. Squares; mature testes, diamonds; pubertal samples in transition to mature morphology, spheres; immature testes. Ellipses delineate estimated 95% confidence intervals for each cluster. Expression is measured using RT-ddPCR.

mature testes, containing mainly mitotic cell types, and in ejaculate, containing mainly mature sperm, compared to immature testes matched for sampling time-point (58 copies/ng (stripped testes) versus 128 copies/ng i.e. 70 copies/ng fewer, 95% CI [30–111 fewer], two-tailed *t*-test, df = 21.8, P = 0.002, $n_{stripped}$ = 8; 19 copies/ng (ejaculate) versus 134 copies/ng i.e. 115 copies/

ng fewer, 95% CI [50–181 fewer], df = 6.6, $P$ = 0.005, $n_{ejaculate}$ = 4, respectively, Fig 2B). Taken together, these results indicate that in salmon gonad tissues, *vgll3* expression is high in immature testes and down-regulated upon the onset of male puberty.

To further test whether *vgll3* down-regulation coincides with the activation of the male puberty pathway, we compared *vgll3* expression relative to the expression of *amh*, which is a repressor of male puberty and Sertoli cell differentiation, and *igf3*, which promotes spermatogonial germ cell differentiation [31–34]. Consistent with our visual classification of gonad morphology, immature testes were characterized by typically higher *amh* expression compared to *igf3* expression, whereas morphologically mature testes with differentiated (running) milt showed typically equal, or higher expression of *igf3* compared to *amh* (S5 Fig). Clustering based on *amh/igf3* expression ratio and *vgll3* expression level further revealed that mature testes were separated in gene expression space from immature testes (bootstrap LRT [35] $P<0.0001$) and characterized by both low *amh/igf3* ratio and low *vgll3* expression, indicating that down-regulation of both *amh* and *vgll3* expression is commonly associated with completing puberty (Fig 2C). In sum, these results point towards a mechanism by which male puberty is associated with a down-regulation of *vgll3* expression, corroborating previous studies that suggested *vgll3* influences Sertoli cell differentiation [27].

While the effects of *vgll3* down-regulation on the downstream transcriptional landscape of testes cell types remain largely unknown, transcriptomic studies indicate that *vgll3* expression correlates with changes in the Hippo signaling pathway, known to control cell differentiation and organ size [27,28]. To gain additional insight into the expression patterns of key players of Hippo signaling pathway and *vgll3* during male puberty, we re-analyzed RNA-sequencing data from immature, prepubertal and pubertal testes from an independent dataset [34]. We found that while maturation correlates with a reduction in *vgll3* expression, the Hippo pathway antagonist for *vgll3*, *yap1* [26,36,37], showed the opposite expression pattern with increasing expression as maturation proceeded (S6A and S6B Fig). *Vgll3* was co-expressed within a group of 7073 other genes out of which 5670 were assigned function in the *NCBI RefSeq* database. Gene Ontology analysis revealed an over-representation in "*transcriptional coactivator function*" in the *vgll3* co-expressed cluster (GO:0003713, $P$ = 0.049, S6C Fig), which included multiple histone acetyltransferases (LOC106607167, LOC106580491, LOC106585100), nuclear receptor coactivators (LOC106607631, LOC100196047, LOC106571447, LOC106579720, LOC106590658), CREB-binding proteins (LOC106590442, LOC106564464) and transcriptional adapter proteins (LOC106581187, LOC106577491), suggesting that *vgll3* is embedded in a larger gene regulatory network potentially influencing gene expression on a transcriptomic level.

## Complex transcript expression differences between vgll3 genotypes

The difference in *vgll3* expression level between mature and immature testes led us to hypothesize that the observed higher probability of maturation in *vgll3*\**EE* males may be linked to differences in *vgll3* expression in immature testes of *vgll3*\**EE* and *vgll3*\**LL* males, thus resulting in differing maturation probabilities for males with alternative genotypes. We tested this hypothesis by comparing the testicular expression from immature individuals with different *vgll3* genotypes.

Using three different RT-ddPCR assays including two that tagged transcribed SNPs within the *vgll3* coding sequence, we observed a complex pattern of expression differences where association with *vgll3* genotype varied between different exons of the *vgll3* transcript. Using an assay spanning the boundary of exon 1 and 2, we observed no expression difference between *vgll3* genotypes ($P$ = 0.564, Fig 3), whereas an assay spanning an SNP in exon 2 showed a 14%

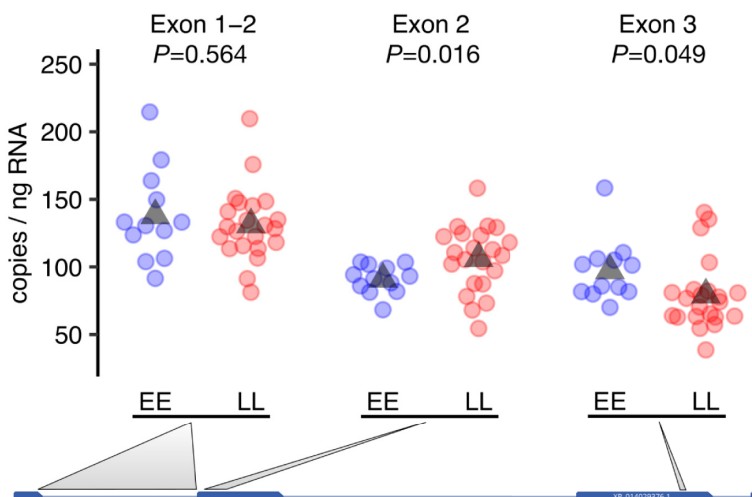

**Fig 3. *Vgll3* expression differences in immature testes using three RT-ddPCR assays in different *vgll3* exons.** Individuals with *vgll3*EE* (blue) and *vgll3*LL* (red) genotypes show varying levels of expression differences along the *vgll3* transcript. Points represent the mean of two replicates per individual sample (triangles; genotype means, *P*-values; two-sided *t*-test). Grey lines connect RT-ddPCR assays with the approximate locations of their amplification targets on the NCBI annotated *vgll3* transcript model with lines representing introns, rectangles representing exons.

lower expression in *vgll3*EE* genotypes compared to *vgll3*LL* genotypes (91 versus 106 copies/ng i.e. 15 copies/ng fewer, 95% CI [3–28 fewer], two-sided *t*-test, df = 31, $P = 0.014$, $n_{EE} = 12$ $n_{LL} = 22$, Fig 3). The opposite pattern was observed using an assay spanning an SNP in exon 3, where *vgll3*EE* genotypes showed 18.5% higher expression compared to *vgll3*LL* genotypes (97 versus 79 copies/ng i.e. 18 copies/ng more, 95% CI [0.08–36 more], two-sided *t*-test, df = 25, $P = 0.049$, $n_{EE} = 13$ $n_{LL} = 23$, Fig 3). The other two tissues with highest *vgll3* expression levels, heart and adipose, did not show a clear relationship between *vgll3* expression level and genotype, maturation or age as assayed on the exon 2 assay (S7 Fig and S8 Fig).

## Rare *vgll3* transcript isoforms in immature testes

The results from RT-ddPCR assays spanning three exons indicated that expression differences between *vgll3* genotypes are restricted to specific exons of the *vgll3* transcript. This suggests that *vgll3* genotype may influence the structure of *vgll3* transcript being expressed, because conflicting expression differences within a gene can be produced by confounding signal from multiple transcript isoforms. We tested this by characterizing *vgll3* transcript structure in immature testes using 5' and 3' RACE (Rapid Amplification of cDNA Ends). Using pooled samples of five individuals for each of the *vgll3*EE*, *vgll3*EL* and *vgll3*LL* genotypes, we observed that all genotypes expressed multiple isoforms that differed in length 5' of exon 3 (S9 Fig). We Sanger sequenced of a total of 278 clones to determine the structure of the expressed isoforms in *vgll3*EL* genotypes and observed at least six distinct RACE fragments representing at least two distinct isoforms (from a total of 202 quality sequences, Fig 4A). The longer isoform aligned with the NCBI annotated transcript with multiple RACE fragments representing alternative lengths of the 5' UTR or exon 1. The shorter second isoform was transcribed from an alternative 5' UTR within the NCBI annotated first intron with translation likely initiated from start codons within exon 2. Further, Sanger sequencing revealed a previously undetected C/T SNP in the alternative 5' UTR at position ssa25:28,655,795, where the genotype was linked to the *vgll3*E/vgll3*L* genotype.

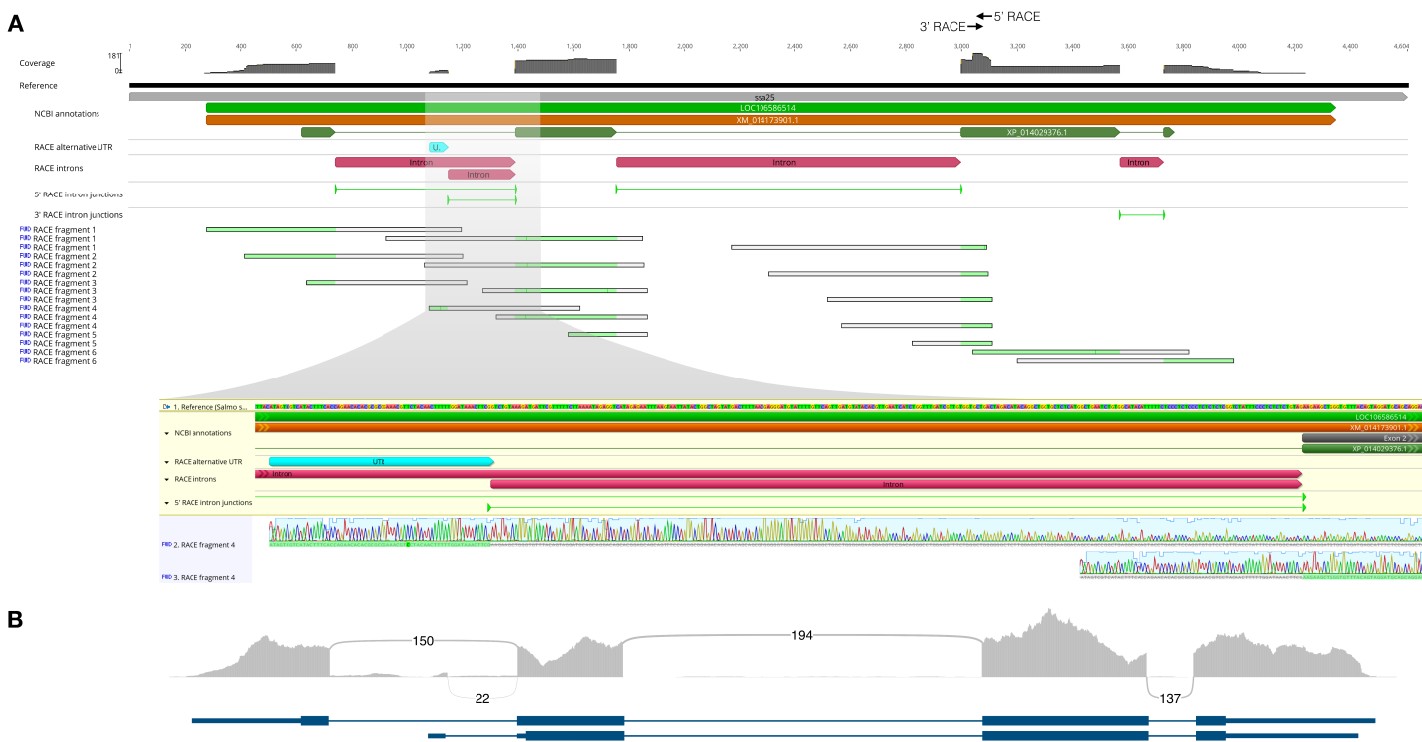

**Fig 4. Characterization of a *vgll3* transcript isoform with an alternative 5' UTR.** (A) 5' and 3' RACE sequencing of pooled total RNA from *vgll3*EL* individuals identified an isoform with an alternative 5' UTR. Browser tracks of Sanger sequencing show (from top) coverage, NCBI annotations, identified UTR, introns and intron junctions, and representative RACE fragments with their alignment to the reference gene model (in light green). Highlighted region shows gene annotations and Sanger sequencing tracks of a representative RACE fragment corresponding to the alternative 5' UTR containing isoform. Alignment to reference is highlighted in green. (B) RNA-seq sequencing coverage and splice events from combined immature, pre-pubertal and pubertal testes. Numbers correspond to RNA-seq reads spanning intron splice sites. Gene models depict transcripts identified using Cufflinks [38].

To further confirm the expression levels of *vgll3* isoforms, we re-analyzed publicly available RNA-seq data from male gonads [34] using Cufflinks [38]. Combined RNA-seq data from three immature, three prepubertal and three pubertal individuals that we identified as *vgll3*EL* heterozygotes supported our discovery of *vgll3* isoforms with alternative transcription start sites. Twenty-two RNA-seq reads mapped to an un-annotated region within intron 1 of the NCBI annotated transcript that spliced into exon 2 with a canonical GT/AG 5'/3' (donor/acceptor) sequence at the splice site (Fig 4B), and that corresponded to the alternative 5' UTR identified above. The abundance of the shorter isoform accounted for 18% of the expression of the sum of *vgll3* isoforms (Cufflinks FPKM 0.56 out of total of FPKM 0.56 + 2.52).

Taken together, based on varying expression across the exons of the *vgll3* transcript in conjunction with the detection of multiple *vgll3* isoforms, we hypothesize that fish carrying *vgll3*EE* and *vgll3*LL* genotypes may preferentially express alternative isoforms of *vgll3* that have different transcription start sites, 5' UTRs and N-terminal protein sequences.

## Cis-regulatory differences in *vgll3* track alternative isoforms

Expression of alternative transcript isoforms can be mediated by *cis* and/or *trans*-regulatory mechanisms, where expression of isoforms is influenced by regulatory differences linked or unlinked, respectively, to the genotype at the locus. We hypothesized that if the expression differences observed between homozygote *vgll3*EE* and *vgll3*LL* individuals were due to *cis*-mediated isoform expression differences, the differing expression levels in homozygotes would

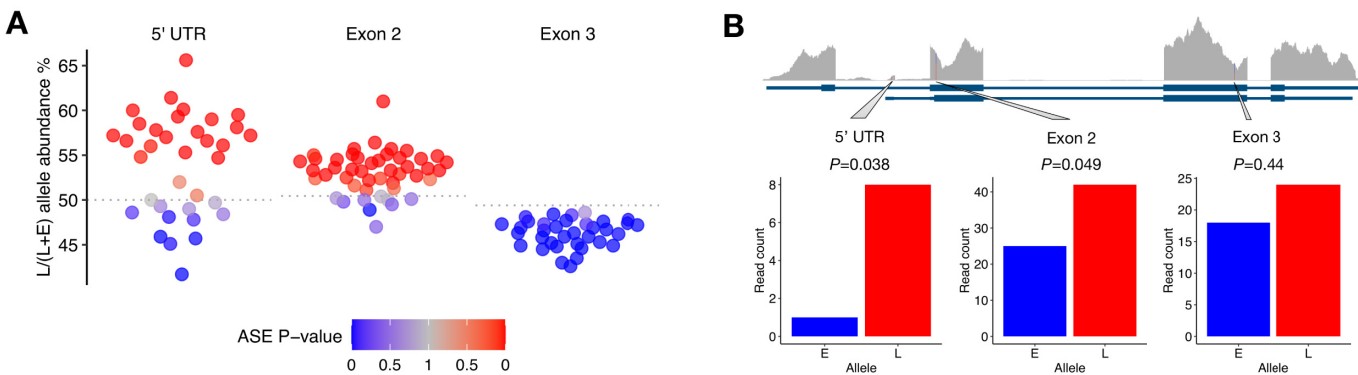

**Fig 5. Allele-specific expression of transcribed SNPs supports a *cis*-regulatory basis for differences in *vgll3* isoform expression.** (A) RT-ddPCR assays of *vgll3* allele-specific expression in heterozygotes show that alternative homozygote expression differences are conserved within heterozygotes. For the alternative 5' UTR, exon 2 and exon 3, on average 54%, 53% and 46% of *vgll3* mRNA molecules are derived from the *late* allele, respectively. For exon 2 and exon 3 assay the null hypothesis for testing of 1:1 allele-specific expression ratio is calculated from heterozygote DNA (dashed line, *N* = 10). The color scale indicates significant (blue; *vgll3*∗*E* allele more expressed, red; *vgll3*∗*L* allele more expressed) and non-significant (grey) allele-specific expression *P*-values (estimated from Poisson modelling of positive RT-ddPCR droplets). (B) Re-analysis of an independent RNA-seq data set is consistent with the RT-ddPCR results and shows allele-specific expression of an alternative 5' UTR and exon 2. Tracks represent (from top) RNA-seq coverage, identified transcript models, and RNA-seq read coverage of SNPs. *P*-value are computed using a binomial test.

be conserved as allele-specific expression within *vgll3*∗*EL* heterozygotes. Expression level differences between the alleles in heterozygotes can be accredited to differences in *cis*-regulation because both alleles are being expressed in the same cellular *trans*-acting environment [39].

Using allele-specific assays for the newly discovered 5' UTR, exon 2 and exon 3, we observed that heterozygotes showed allele-specific expression differences consistent with a largely *cis*-regulatory basis of isoform expression differences. For the alternative 5' UTR, heterozygotes tended to express the *vgll3*∗*L* allele at a higher level compared to the *vgll3*∗*E* allele, with on average 54% of mRNA molecules representing from the *vgll3*∗*L* allele. Using Poisson modeling of expression rates, 21 of the 35 tested heterozygotes (60%) showed significantly higher *vgll3*∗*L* expression at *P*-value threshold of 0.05, while only five (14%) showed higher *vgll3*∗*E* allele expression at the same *P*-value threshold (Fig 5A).

For exon 2, where we observed higher expression in *vgll3*∗*LL* homozygotes compared to *vgll3*∗*EE* individuals, allele-specific expression showed a tendency towards higher expression of the *vgll3*∗*L* allele with on average 53% of the *vgll3* mRNA molecules expressed from the *vgll3*∗*L* allele (Fig 5A), versus 50.4% expected from heterozygous DNA used as control for differences in efficiencies of allele-specific probes. Thirty-one of the 47 heterozygotes analyzed (66%) showed significantly higher *vgll3*∗*L* allele expression, while only one individual (2%) showed significantly higher *vgll3*∗*E* allele expression (*P*<0.05, Poisson rate-ratio test, Fig 5A). In terms of expressed mRNA molecules, the average magnitude of the *cis*-regulatory differences observed between *vgll3*∗*E* and *vgll3*∗*L* alleles (6.8 mRNA molecules per ng of RNA) was large enough to account for 89% of the average expression differences between *vgll3*∗*EE* and *vgll3*∗*LL* homozygotes (15.3 mRNA molecules, i.e. (2∗6.8)/15.3∗100 = 89%).

For exon 3, where *vgll3*∗*LL* homozygotes showed lower expression compared to *vgll3*∗*EE*, allele-specific expression differences were also consistent with *cis*-regulation with on average 46.2% of mRNA molecules expressed from the *vgll3*∗*L* allele in *vgll3*∗*EL* heterozygotes (compared to 49.4% in heterozygous DNA control). Thirty-one of 35 tested heterozygotes (89%) showed significantly lower expression of the *vgll3*∗*L* allele (*P*<0.05). The average magnitude of the *cis*-regulatory difference (7.5 mRNA molecules per ng of RNA) accounted for 83% of the average homozygote expression difference (18 mRNA molecules, i.e. (2∗7.5)/18∗100 = 83%).

For further validation of the *cis*-regulatory differences in *vgll3*, we re-analyzed publicly available RNA-seq data from immature male gonads [34]. We identified three immature individuals as heterozygous for the same *vgll3* SNPs we analyzed with our allele-specific RT-ddPCR assays and quantified *vgll3* allele-specific expression based on RNA-seq reads overlapping the SNP positions. Inspection of RNA-seq reads revealed the individuals to be heterozygous for all of the three linked SNPs in *vgll3* targeted by our RT-ddPCR assays (Fig 5B). Consistent with our results from RT-ddPCR, the newly discovered alternative 5' UTR showed strong allele-specific expression, being expressed almost solely from the *vgll3*L allele when reads from all three males were combined (8 versus 1 reads, binomial exact test $P = 0.038$). The *vgll3*L allele in exon 2 had higher expression compared to the *vgll3*E allele in all three heterozygote individuals (13 versus 8 reads, 10 versus 5 reads, 11 versus 10 reads, combined 42 reads versus 25 reads, binomial exact test $P = 0.049$, Fig 5B). For exon 3, the difference in *vgll3*L and *vgll3*E alleles was non-significant ($P = 0.44$).

These results show that the *vgll3* allele expression differences observed between alternative homozygotes tend to be conserved in direction and in magnitude when assayed in heterozygotes. Re-analysis of an independent data-set supports this conclusion. In sum, our results strongly support that the differences in maturation timing between *vgll3* genotypes is being driven by *cis*-regulated expression differences of alternative *vgll3* isoforms.

## Discussion

The molecular basis of variation in life history traits and their trade-offs have thus far remained elusive because many life history traits are either polygenic, experimentally intractable, or their genetic basis has only recently been discovered. Here, we show that variation in age at maturity, a key life history trait, is associated with *cis*-regulatory differences in isoform expression of the transcription co-factor gene *vgll3* in immature testes of Atlantic salmon males. Using targeted expression assays, time series sampling in ten tissues and characterization of transcript structure, we discovered that genotypes that tend to mature early versus those that tend to mature late express different mixtures of *vgll3* transcript isoforms pre-puberty. The expression differences were allele-specific in heterozygotes, suggesting that divergence in *cis*-regulation underlies the associations between *vgll3* genotype, isoform expression and maturation probability.

Our results provide evidence to suggest that *vgll3* genotypes associated with maturation age phenotypes may do so through a mechanism of *cis*-regulated differences in transcript isoform expression in immature testes. The expressed *vgll3* isoforms differed in their transcription start sites, but are predicted to all produce a functional *vgll3* protein. Alternative transcription start sites are thought to be important for determining cell-types and their proliferation by influencing not only protein sequence, but also through post-transcriptional regulation such as mRNA translation and stability [22,40,41]. Validating the functional differences between *vgll3* isoforms and characterizing isoform expression patterns of *vgll3* in additional tissues will be an important avenue for future research. Additionally, the opposite direction of association between *vgll3* genotype and *cis*-regulatory expression in exons 2 and 3 indicates the possible existence of additional transcript isoforms. Expression of different transcript isoforms has been established as the primary mechanism of important developmental turning points in several other species. In many insects for example, alternatively spliced isoforms of the gene *doublesex* act as sex determining factors [42]. Most species-specific differences in transcript isoform expression have been linked to *cis*-regulatory differences [43,44]. *Cis*-regulatory changes that affect alternative splicing are thought to reside within the un-spliced pre-mRNA [44]. For expression from alternative transcription start sites such conservation of *cis*-

regulatory placement has not been shown, as initiation of transcription can be influenced by regulatory elements not only in the core promoter, but also in distal enhancer sites [45]. Overall, genetic variation influencing *cis*-regulatory loci is known to be an important source of phenotypic diversity and a common source of evolutionary change [46–52]. The results reported here support the notion that this is also a strong candidate for the molecular mechanisms underlying variation in complex life histories as well.

The transcription co-factor function of *vgll3* suggests that isoform changes in *vgll3* expression may transmit downstream to expression changes in gene regulatory networks associated with cell proliferation and organ development [37]. Like its vertebrate and invertebrate homologs, the salmon *vgll3* has been proposed to control cell differentiation through interacting with the TEAD-family of transcription factors of the Hippo signaling pathway [27,28]. Vgll3, like related co-factors in mice, humans and *Drosophila*, is predicted to compete for TEAD accessibility with alternative co-factors downstream of the Hippo signaling pathway controlling for organ size (Yap/Taz co-factors) [26,36]. Consistent with this, by analyzing RNA-sequencing data from immature, prepubertal and pubertal testes, we found that *vgll3* and *yap1* showed opposite expression patterns with respect to maturity status. Salmon *vgll3* expression was also found to negatively correlate with *yap1* expression in early development [28]. We hypothesize that changes in *vgll3* isoform abundance associated with *vgll3*\*E and *vgll3*\*L alleles may influence the tendency of cell differentiation in immature testes by changing interaction between Vgll3 and TEADs and/or Vgll3-Yap (Fig 6). Clarifying the molecular relationships between *vgll3*, TEADs and *yap1* remain as important topics for future research.

The relatively low expression level of *vgll3* and variation around genotype means also merit careful consideration of the limits of our study. First, some of the variation around the mean expression levels within *vgll3* genotypes can possibly be explained by additional genetic variation segregating in the experimental families and influencing *vgll3* expression in *cis* and in *trans*. Bimodality in the expression of *vgll3* 5'UTR (Fig 5A) suggests that the influence of as yet unknown *trans* acting factors on *vgll3* isoform expression cannot be ruled out. Second, as the overall expression level of *vgll3* is very low, an alternative interpretation of the results could be that the small but statistically significant differences in expression across different exons of the *vgll3* transcript are driven by non-biological sources, such as noise in the measurement of low expression levels, or differences in assay efficiency and specificity. However, concurring results from re-analysis of an independent data-set, our careful control experiments, and the fact that the observed expression differences were specific to tissues and not observed in a systematic manner across all samples do not support this interpretation. The possibility exists also that *vgll3* is included in, but not determining the output of, the molecular pathway determining maturation timing, or that variation in the timing of measurement versus initiation of puberty causes variation in the genotype-phenotype association. Further functional validation through manipulation of isoform expression and allele-specific expression is an important future direction for validating the role of *vgll3* isoform-specific *cis*-regulation in maturation timing.

It is important to note that although we observed a *cis*-regulatory differences associated with isoform expression and maturation probability, it does not exclude the possibility that protein-coding changes within the *vgll3* gene [4,5] could play additional functional roles in determining maturation timing. Further, inferring an expression-genotype-phenotype association in male gonads does not exclude the possibility that *vgll3* also has functional relevance in other tissues/processes (e.g. adipogenesis). In fact, pleiotropic effects of single genes that are under the control of tissue and development-specific regulatory sequences fits well with what we understand from the modularity of *cis*-regulation [46]. The expression pattern of *vgll3* and its homeolog indeed support additional functions for the gene pair in adipogenesis and female gonad development [28]. In addition, we observed relatively high *vgll3* expression in heart as

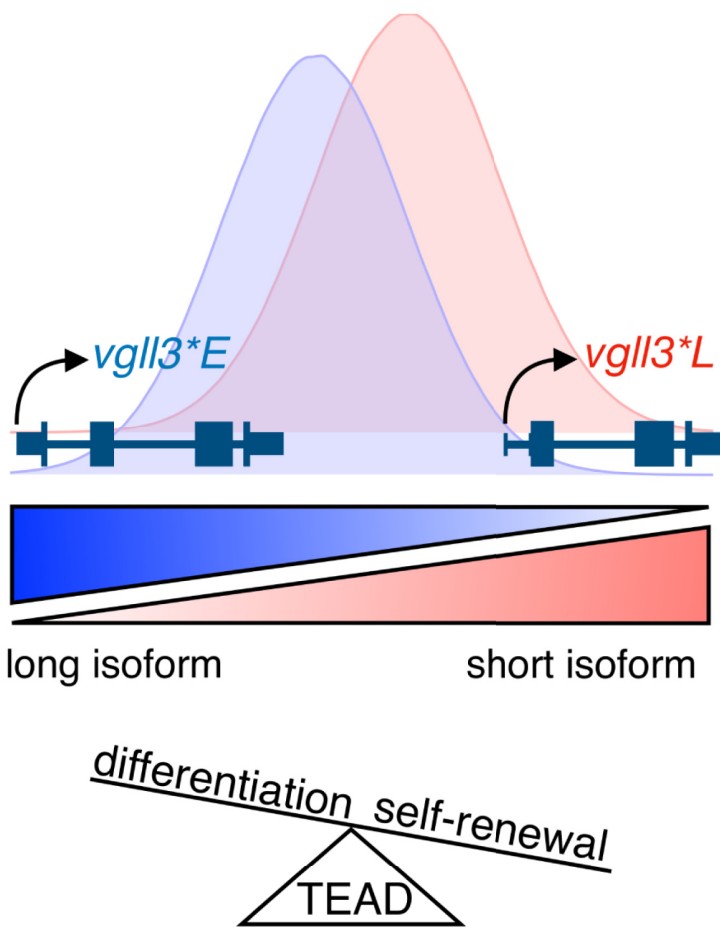

**Fig 6. Model for a molecular mechanism of *vgll3* genotype-dependent liability in puberty timing.** Hypothetical distributions represent relative abundances of *vgll3* isoforms in individuals with given *vgll3* genotypes (blue, *vgll3*\*E*; red, *vgll3*\*L*). We hypothesize that *cis*-regulatory differences influencing *vgll3* isoform expression are an important mechanism contributing to the association between *vgll3* genotype and maturation timing. The *vgll3*\*L* allele tends to express more of the shorter isoform compared to the *vgll3*\*E* allele, thus potentially changing the balance between *vgll3*-controlled differentiation versus self-renewal. TEAD interaction with *vgll3* isoforms may tip the balance between tendency for self-renewal (higher expression of the short isoform) and differentiation (long isoform).

well as testes, which may indicate a pleiotropic action of *vgll3* in heart development and function through the Hippo signaling pathway [53]. The role of changes in isoform expression of *vgll3* in such pleiotropic effects warrants future research.

Although the effects of protein-coding changes on maturation phenotype were not addressed here, support for the benefits of *cis*-regulatory changes compared to protein-coding changes in the evolution of phenotypic diversity can be drawn from studies on morphological traits. Regulatory changes often show higher modularity across tissues and development, additivity, and lower pleiotropy, which are thought to inflict less evolutionary constraints compared to protein-coding changes [46,54], especially in cases where proteins modulate the transcription of other genes as is the case with *vgll3*. These features of evolvability of regulation are also well demonstrated here, where we observed alternative expression dynamics (e.g. rising versus diminishing expression in heart and liver, respectively, as individuals age), and genetic effects driving expression differences restricted to one tissue and developmental stage

(immature testes). Overall, these features are expected to lead to *cis*-regulation being a more likely source of evolution in life history traits compared to coding changes.

We have shown here that polymorphisms in the *vgll3* region associate significantly with one-year-old male age at maturity in controlled conditions. The effect of *vgll3* genotype on maturation timing in our experiment was arguably smaller compared to that observed in wild populations and for sea-age at maturity [4]. Possible explanations for the divergent finding in the two studies can be related to the differences in (*i*) traits (age at maturity versus probability of maturation), (*ii*) modeling approaches (based on logit versus in our case probit distributions), (*iii*) wild versus husbandry environments, (*iv*) differences in the assessment of maturity status, (*v*) variation in *vgll3* contribution to age at maturity phenotypes across populations and (*vi*) possible differences in the genetic architecture of the distinct age at maturity phenotypes (sea-age at maturity versus early male maturity).

Taken together, here we have revealed key functional insights into the temporal expression dynamics and likely molecular mechanisms that translate genetic variation in a major effect locus to variation in life history strategies in Atlantic salmon. Many life history traits are encoded by many genes where it has remained challenging to identify the loci responsible for variation [55]. In the light of the results presented here, gene regulatory evolution and isoform variation are likely candidates for mechanisms mediating alternative life history strategies that merits further investigation across species.

## Materials and methods

### Ethics statement

Experimentation was conducted according to the Finnish Government Decree on the Protection of Animals Used for Scientific or Educational Purposes (564/2013), which implements EU directive 2010/63/EU. The experiments in this study were approved by the Project Authorisation Board (ELLA) on behalf of the Regional Administrative Agency for Southern Finland (ESAVI) under experimental license ESAVI/2778/2018.

### Crosses used in the study

We controlled-crossed eight 2x2 factorial matings of unrelated individuals that each included one *vgll3*EE* and one *vgll3*LL* male and female (see ref [30] for genotyping details), thereby resulting in 32 families with reciprocal *vgll3* genotypes *EE*, *EL*, *LE* and *LL* (allele order denotes female and male parents of origin respectively, S1 Fig). We used parents from the "Oulujoki" broodstock, which is a mixture of several salmon populations from northern Baltic rivers and is maintained by the Natural Resources Institute Finland (LUKE) [56]. All parents were selected for having a homogenous age-at-maturity-associated haplotype encompassing the *vgll3* (chromosome 25 homeolog) coding sequence, the top non-coding association [4], and the coding region of the adjacent *akap11* gene (collectively referred to as "*vgll3* genotype").

### Husbandry design and conditions

Following fertilization in October 2017, familial egg batch replicates were raised in two egg incubators until first feeding (~ 2 months post-hatching) with mesh-separators to contain families in individual compartments. Families were assigned to compartments randomly. At first feeding, familial batches were combined and 48 individuals per family were randomly selected and distributed evenly across eight 0.25 m$^3$ tanks using recirculated water i.e. on average, each tank included six individuals per family. Prior to tank allocation, each individual was tagged with visible elastomer (VIE) at the base of the caudal fin to enable identification of the *vgll3*

genotype of an individual by a fluorescent color (initial $N$ = 1300). Light cycle and water temperature (min = 6.3, max = 17.7˚C) during husbandry corresponded to the local seasonal cycle at 61˚N. Fish were fed first with live *Artemia* for three days, after which we fed commercial aquaculture feed *ad libitum* (Raisio Baltic Blend) with pellet diameter increasing over time according to fish size. Temperature, oxygen and nitrogen-component levels were monitored on a regular basis.

## Individual tagging, family assignment and morphological measurements

At eight months post-hatching and before the onset of the first breeding season we tagged remaining fish using passive integrated transponder (PIT) tags and collected fin-clips. We extracted DNA using standard chelex or salt extraction methods, genotyped all fish for 141 informative SNPs [57], assigned their genotypic sex based on genotyping coverage of the salmon sex determining locus *sdy* and assigned individuals to families using the likelihood method implemented in SNPPIT [58]. We recorded the length and mass of all fish monthly after PIT-tagging. We non-lethally assessed the maturity status of all males by testing for the presence of running milt at nine months post-hatching. Morphological measurement and non-lethal maturation assessment were performed in a random order such that the assessor did not have knowledge of *vgll3* genotype.

## Statistical model of male maturation probability

We modeled the probability of males reaching maturity within their first year given their *vgll3* genotypes by using a GLMM (generalized linear mixed model) approach to implement a genetic threshold model that takes full account of the relatedness among individuals [59,60]. We estimated maturation probability using a generalized linear animal model with probit-link function under Bayesian Markov Chain Monte Carlo simulations with the R-software [61] version 3.6.1 and package MCMCglmm [62] for the following model: $y = vgll3 + length + a + t + e$, where y is a vector of maturation binaries, *length* is mean-centered and variance scaled natural logarithm of length (cm), *vgll3* is the genotype, $a$ is the additive genetic relationship-matrix predicted animal effect [63], $t$ is the common environmental tank effect and $e$ the error term (with fixed variance of 1). We used priors following a $\chi_1^2$ distribution [64] and based inferences on 10,000 samples collected every 1,000 iteration after burn-in of 10,000. Model convergence and adequate sampling was ensured by visual trace plot inspection, autocorrelation < 0.1 for lag-2 parameter samples, and using Heidelberger and Welch's convergence diagnostic [65]. We transformed maturation probabilities from probit-scale to normal probability scale for ease of interpretation. Heterozygote *vgll3*LE* and *vgll3*EL* (order of alleles denotes maternal origin) were considered as one group as initial analysis indicated that maturation probabilities between these genotypes were not significantly different.

## Temporal sampling, RNA extraction and RT-ddPCR

We sampled individuals with *vgll3*EE*, *vgll3*EL*, *vgll3*LE* and *vgll3*LL* monthly (3–32 in total per genotype per time point) and assessed sex, gonad size and maturity by dissection. We sacrificed the animals by MS-222 overdose and dissected ten tissues (brain (timepoints 12–13 months post hatch); heart, liver, muscle (timepoints 3–13 months post hatch), adipose (timepoints 5–13 months post hatch); immature ovary (timepoints 6–13 months post hatch); immature testes (timepoints 7–13 months post hatch); mature testes (timepoints 8 and 10–12 months post hatch); stripped mature testes (timepoints 12–13 months post hatch) and ejaculate (timepoint 13 months post hatch)). As immature male gonads had no recordable weight on a scale with a detection limit of 10 mg thus preventing the use of gonadal-somatic index, we

classified each sacrificed male fish visually as immature (not enlarged, transparent color gonads), pubertal (slightly enlarged, transparent or white color gonads) or mature (enlarged white color gonads). We flash froze the samples on liquid nitrogen and stored the samples in -80˚C. We homogenized the tissues using an OMNI Bead Ruptor Elite machine, in 2 ml or 7 ml (for mature gonads) with 2.4 mm steel beads. We used Macherey-Nagel NucleoSpin RNA kits in single and in 96-well plate formats to extract total RNA and treated the samples with additional DNase (Invitrogen Turbo Dnase) to remove any residual DNA. We verified total RNA quality using Agilent BioAnalyzer for a random set of 24 samples and found all samples to have RIN>9. We measured total RNA yield using ThermoFischer Qubit and Quant-it reagents, diluted the samples to working concentrations and re-measured RNA concentrations. For samples in Figs 2 and 3 (all gonad samples) we repeated the RNA concentration measurement once more and calculated the mean of the duplicate measurements, which we used in data normalization. We used ~20 ng (2–8 months post-hatch) or ~50 ng (8–14 months post-hatch) of total RNA as template in one-step reverse transcription droplet digital PCR (RT-ddPCR) [66] (BioRad) with TaqMan probes specific to *vgll3* alleles to quantify *vgll3* absolute expression (S1 Table). Measurement of *amh* and *igf3* expression was performed as above, with the exception of using ~2 ng of total RNA template and TaqMan probes that were labeled with alternative fluorophores and assayed in the same RT-ddPCR reactions. We verified the specificity of *vgll3*-genotype specific TaqMan assays with genomic and synthetic DNA templates (S10 Fig) and optimized melting temperatures for all assays using a gradient. Genomic DNA could not be used as control for the *vgll3* 5' UTR assay because the assay spans an intron-exon boundary. Genomic and synthetic DNA controls and null control were included in each RT-ddPCR *vgll3* run, while genomic DNA and null control were included in *Amh* and *Igf3* RT-ddPCR runs.

## Expression analyses

We analyzed RT-ddPCR results using BioRad QuantaSoft software. We manually set minimum signal thresholds for *vgll3*\*L and *vgll3*\*E channels because visual inspection of RT-ddPCR signal indicated that, for *vgll3* exon 2 assay, the *vgll3*\*L specific allele showed low background signal on the *vgll3*\*E channel (S10 Fig). We excluded samples that failed *vgll3* RT-ddPCR reaction using a pre-determined minimum threshold of 10,000 quality-filter passed droplets to include only samples where expression could be estimated with low error. We excluded *amh*/*igf3* expression data from samples where RT-ddPCR signal was saturated (over 98% of droplets were positive) because mRNA abundances cannot be accurately estimated with saturated RT-ddPCR data. We excluded seven samples where the genotypic sex was inconsistent with morphological sex of the gonads. We excluded three mis-labeled individuals with inconsistent *vgll3* genotypes and allelic expression (e.g. *vgll3*\*EE homozygotes with expression signal only from *vgll3*\*LL specific probes). We removed two heterozygote samples that showed extreme monoallelic expression of the *vgll3*\*L allele (fractional abundance over 90%), possibly representing labeling or sampling error. We then used custom R scripts to normalize the number of *vgll3* mRNA molecules to the amount of input RNA and analyze the data and plot the results (Dryad doi: 10.5061/dryad.k6djh9w4w).

For investigation of the effect of age/season and *vgll3* genotype to *vgll3* expression in immature testes and immature ovaries, we used the following regression model: $y = vgll3 + x + e$, where $y$ is *vgll3* total mRNA copies, $x$ is days post-hatch, *vgll3* is the genotype and $e$ the error term. We tested the effect of *vgll3* and $x$ (days post-hatch) individually using a corresponding model.

We characterized the relative expression levels of the key male maturity genes *amh* and *igf3* by way of Poisson modeling of expression rate-ratio for each sample. The number of positive droplets with expression signal in RT-ddPCR data follows a Poisson distribution with lambda being the rate of positive droplets over all droplets ([66], https://www.bio-rad.com/webroot/web/pdf/lsr/literature/Bulletin_6407.pdf). We calculated the Poisson rates of expression from *amh* and *igf3* -positive droplets over the total amount of analyzed droplets (the rate ratio of *amh/igf3* expression) using the R function *poisson.test*.

We tested for presence of distinct clusters of gene expression patterns on *amh/igf3* and *vgll3* expression space using Gaussian mixture modeling as implemented in the R package *Mclust* version 5 [35]. We chose the best fitting mixture of Gaussian distributions based on Bayesian Information Criterion using the function *mclustBIC* and further tested the best fit mixture model for the presence of more than one distributions using a bootstrapping likelihood-ratio test as implemented in the function *mclustBootstrapLRT* with 1000 bootstrap replications.

## RACE

To perform RACE we followed the manufacturer's instructions using Takara SMARTer 5'/3' RACE-kit with the following modifications. We pooled equal amounts (25 ng) of immature testis total RNA from six individuals per genotype with either *vgll3*EE*, *vgll3*EL* or *vgll3*LL* genotypes to create three genotype specific pools, and performed RACE with primers aligning to exon 3 of the *vgll3* transcript. After analysis on agarose gel, we purified the RACE PCR products using Macherey-Nagel NucleoMag DNA binding beads, cloned the PCR products to a pRACE vector using the SMARTer kit reagents, transformed the plasmids to *E. coli* and grew the transformation at 37 degrees C o/n on LB-ampicillin (100 mg/ml) plates. Sanger sequencing of transformed PCR products was performed as described in ref [67]. Briefly, colonies were picked into 96 deep-well plates containing 1 ml growth media and incubated in a shaker o/n. An aliquot (150 ul) was transfered to a PCR plate and spin down. The pellet was dissolved in 150 water and heated to 95 C for 10 min and 2 ul was used as template in PCR with universal and reverse primers (UP-47 and RP-48). The obtained PCR products were sequenced with UP and RP universal primers using BigDye Chemistry v 3 and analyzed on ABI 3130xl Sequencer (Life Technologies) at the DNA Sequencing and Genomics lab Institute of Biotechnology University of Helsinki. We used *Geneious Prime 2020.1.2* to automatically detect and trim the sequences of vector DNA and low-quality sequences above 1% base error rate. We aligned the sequences to the *vgll3* region downloaded from NCBI using the *Geneious RNA* mapper with the highest sensitivity settings and allowing for novel intron detection.

## Allele-specific expression

Testing of *vgll3* allele-specific expression in heterozygotes was performed in a similar manner as above for *amh/igf3* expression (test for Poisson rate ratio), with the exception that signal from positive droplets now represented expression from either the *vgll3*E* or the *vgll3*L* allele. We defined the expression rate of *vgll3*E* allele as the number of droplets positive for the *vgll3*E* probes over all probes that passes the QuantaSoft standard quality filter (including empty droplets), and correspondingly for the *vgll3*L* allele. We then tested for difference in Poisson rates for each individual heterozygote using the R function *poisson.test* and *P*-value threshold of 0.05. We calculated the null expectation for equal expression rates based on RT-ddPCR with 10 replicates of *vgll3* heterozygous DNA as template. The assay for the alternative 5' UTR spanned an intron-exon boundary, and therefore we could not use genomic DNA as control (the assay does not amplify with DNA as template). We assumed 50% efficiency for this assay.

We estimated that on average the *cis*-regulatory difference in *vgll3* allele-specific expression can account for 89% of the average expression difference between *vgll3*EE* and *vgll3*LL* homozygotes in exon 2 and 83% in exon 3. We estimated the 89% for exon 2 as follows (analogous calculation for exon 3 not shown): On average *vgll3*EL* individuals expressed 113 *vgll3* mRNA molecules (per ng of input RNA, Fig 3A). Of these, on average, 53.5 molecules were expressed from the *vgll3*E* allele and 60.3 molecules from the *vgll3*L* allele, giving an average *vgll3*L* relative abundance of 53% (60.3/(53.5+60.3)*100 = 53%). From above we can see that average difference between *vgll3*E* and *vgll3*L* mRNA counts is 6.8 mRNA molecules (60.3–53.5 = 6.8). Corresponding, average difference between *vgll3*EE* and *vgll3*LL* genotypes is 15.3 mRNA molecules (106.35–91.06 = 15.3). The percentage of *vgll3*EE*- *vgll3*LL* difference accounted by *vgll3*E*- *vgll3*L* cis*-expression difference is thus (2*6.8)/15.3*100 = 89%, as *vgll3*EE* and *vgll3*LL* have two of either allele each.

## RNA-seq analyses

We re-analyzed RNA-sequencing data from immature, prepubertal and pubertal Atlantic salmon testis samples [34] in order to investigate the relative expression patterns of *vgll3* and *yap1* genes. We downloaded raw RNA-seq data for immature (SRR8479243, SRR8479245, SRR8479246), prepubertal (SRR8479244, SRR8479249, SRR8479250) and pubertal (SRR8479242, SRR8479248, SRR8479247) testes from the *Sequence Read Archive* and filtered the reads for quality using *Fastp* and default parameters [68].We created an alignment reference with *STAR* [69] and the parameter *–runMode genomeGenerate*, using the Atlantic salmon reference genome and annotation files downloaded from *NCBI* [70]. We aligned and quantified the quality-filtered reads using *STAR* in manual two-pass mode and with the following parameters:*—outFilterIntronMotifs RemoveNoncanonicalUnannotated–chimSegmentMin 10 – outFilterType BySJout–alignSJDBoverhangMin 1 –alignIntronMin 20 –alignIntronMax 1000000 –alignMatesGapMax 1000000 –quantMode GeneCounts–alignEndsProtrude 10 ConcordantPair—limitOutSJcollapsed 5000000*. We combined the read counts from *STAR* into a single table using a custom *R* script and normalized the data using *DESeq2 varianceStabilizingNormalization* [71] and independently with *Clust* [72] (automatic normalization method: 101 3 4), so that expression patterns of genes could be compared independently from their total expression level. Last, we used custom scripts in *R* to analyze the co-expressed genes and plot the results. *NCBI RefSeq* annotations for genes were fetched using *annotationhub* package and Gene Ontology over-representation in co-expression groups was tested using *clusterProfiler* package [73]. Reads from isoform variants were quantified using *ggsashimi* [74]

We quantified *vgll3* allele-specific expression in the RNA-seq data by comparing reads overlapping SNP positions Ssa25: 28655795, Ssa25:28656101 and Ssa25:28658151 (the two latter being same SNPs used in our crossing scheme and allele-specific TaqMan assays). We changed the SNP nucleotide to "N" in order to remove reference-biased mapping and re-aligned the RNA-seq reads using *STAR*, excluding multimapping reads and reads with more than one mis-match by defining parameters *-outFilterMultimapNmax 1 -outFilterMismatchNmax 1* (in addition to the parameters defined above). We then counted the number of overlapping *vgll3*L* and *vgll3*E* reads by manual inspection of RNA-seq read tracks using *IGV* [75] and tested the significance of allele-specific expression to the null expectation of 1:1 expression with binomial test.

## Supporting information

**S1 Fig. Outline of experimental crosses.** 16 females with 16 males were crossed in a 2x2 factorial design, producing 32 families with *vgll3* genotypes *EE*, *EL*, *LE* and *LL* (order denotes

maternal and paternal origin of alleles).
(EPS)

**S2 Fig. Maturation frequencies at nine months post-hatching.** Number above bars indicates the number of mature individuals over all individuals.
(EPS)

**S3 Fig. *Vgll3* expression in all investigated tissues and whole fry.** Stripped testes; mature testes with ejaculate manually removed.
(EPS)

**S4 Fig. *Vgll3* expression in the liver.** *Vgll3* expression in liver shows a linear downward trend, but the overall expression level is low. Green line, loess smoothing; grey shading, 95% confidence interval.
(EPS)

**S5 Fig. Expression of puberty indicator genes in male testes.** Average *amh/igf3* expression ratio is higher in immature testes and maturation coincides with relative down-regulation of *amh* in relation to *igf3*. Pubertal samples show large variation in *amh/igf3* ratio. Testes are separated into categories based on morphology (see methods).
(EPS)

**S6 Fig. RNA-seq analysis of *vgll3* co-expression.** Re-analysis of data from ref [34] Expression patterns of *vgll3* and *yap1* in immature, prepubertal and pubertal testes. (A) Points represent normalized expression levels (variance stabilizing normalization [71]). (B) Points represent the relative normalized expression level means [72] from three biological replicates in each case. (C) Relative expression patterns of 7074 genes in *vgll3* co-expression cluster. Black points; *vgll3*, grey points; *yap1*.
(EPS)

**S7 Fig. Expression of *vgll3* in the heart.** All trendlines non-significant at *P*-value threshold 0.05.
(EPS)

**S8 Fig. Expression of *vgll3* in adipose tissue.** All trendlines non-significant at *P*-value threshold 0.05.
(EPS)

**S9 Fig. RACE characterization of *vgll3* transcripts.** Agarose gel of 5' and 3' RACE PCR reactions. Both 5' and 3' RACE reactions amplify multiple RACE fragments of different lengths. RACE was performed using primers aligning to exon 3 of the *vgll3* transcript.
(EPS)

**S10 Fig. Genomic DNA and synthetic DNA controls of allele-specific *vgll3* assays.**
(EPS)

**S1 Table. TaqMan assays used in the study.**
(DOCX)

## Acknowledgments

We thank members of Primmer lab for assistance with the common garden experiment, Lars Paulin for assistance in RACE and Sanger sequencing, and LUKE staff for assistance with the initial crosses.

## Author Contributions

**Conceptualization:** Jukka-Pekka Verta, Paul Vincent Debes, Craig Robert Primmer.

**Data curation:** Jukka-Pekka Verta, Paul Vincent Debes.

**Formal analysis:** Jukka-Pekka Verta.

**Funding acquisition:** Craig Robert Primmer.

**Investigation:** Jukka-Pekka Verta, Paul Vincent Debes, Nikolai Piavchenko, Annukka Ruoko-lainen, Outi Ovaskainen, Jacqueline Emmanuel Moustakas-Verho, Seija Tillanen, Noora Parre.

**Methodology:** Jukka-Pekka Verta, Paul Vincent Debes, Tutku Aykanat.

**Project administration:** Jukka-Pekka Verta, Paul Vincent Debes, Craig Robert Primmer.

**Resources:** Jaakko Erkinaro.

**Software:** Jukka-Pekka Verta, Paul Vincent Debes, Tutku Aykanat.

**Supervision:** Jukka-Pekka Verta, Paul Vincent Debes, Craig Robert Primmer.

**Visualization:** Jukka-Pekka Verta.

**Writing – original draft:** Jukka-Pekka Verta.

**Writing – review & editing:** Jukka-Pekka Verta, Paul Vincent Debes, Jacqueline Emmanuel Moustakas-Verho, Craig Robert Primmer.

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
