## [Decision Letter · Decision Letter 0]

16 Jan 2020

Dear Dr Verta,

Thank you very much for submitting your Research Article entitled 'A cis-regulatory mechanism for life history strategy variation in Atlantic salmon' to PLOS Genetics. Your manuscript was fully evaluated at the editorial level and by independent peer reviewers. The reviewers appreciated the attention to an important problem, but raised some substantial concerns about the current manuscript. Based on the reviews, we will not be able to accept this version of the manuscript, but we would be willing to review again a much-revised version. We cannot, of course, promise publication at that time.

If you decide to revise the manuscript for further consideration at PLOS Genetics, please aim to resubmit within the next 60 days, unless it will take extra time to address the concerns of the reviewers, in which case we would appreciate an expected resubmission date by email to plosgenetics@plos.org.

[LINK]

We are sorry that we cannot be more positive about your manuscript at this stage. Please do not hesitate to contact us if you have any concerns or questions.

Yours sincerely,

Juliette de Meaux

Associate Editor

PLOS Genetics

Kirsten Bomblies

Section Editor: Evolution

PLOS Genetics

Dear Authors,

Your manuscript reports the impact of Vgll3, a transcription factor, on variation of the maturity age in atlantic salmon. The relevance of your study finds unanimous support among all three reviewers. Their careful examination of your manuscript has yielded many constructive comments that will help improve the manuscript. As a result, I ask you to major revise your manuscript before it can be accepted for publication. These revisions will have to address each of the comments made by the reviewers. I particularly believe that the description of effect sizes and their justification must be at the center of your attention in the preparation of the revised manuscript. This concern is shared by two of the reviewers. I was personally surprised to see that the quantitative RT PCR you performed did not use control house keeping genes, but were just standardized to the amount of mRNA. This means that variance for the efficiency of cDNA synthesis was not controlled for in your assay. I also believe that the magnitude of the cis-regulatory change you report is very small, which requires a careful discussion of both potential technical limitations and putative relevance of tiny changes in cis-regulatory activity. Vgll3 is also highly expressed in the heart and knowledge regarding its potential pleiotropic action should be discussed in more detail. Finally, I would like to ask you to broaden the discussion of your data for the broader readership of our journal. It seems indeed a bit far fetched to claim in the discussion that your study indicates that the evolution of life history is more likely to be regulatory than coding.

Reviewer's Responses to Questions

**Comments to the Authors:**

Reviewer #1: Please find my comments in the attached pdf file.

Reviewer #2: The paper by Verta et al. addresses the molecular basis of life history differences in Atlantic Salmon. From prior work, they identify a candidate transcription factor, vgII3, that is associated with age at maturity and confirm this association using crosses in the laboratory. A number of factors suggest that changes in vgII3 expression may be important and the authors analyze expression across multiple tissues and time points, identifying a change in vgII3 expression in the testis upon maturity. They then test for regulatory differences among individuals with different vgII3 ‘alleles’ finding a small, but significant cis-regulatory difference in expression prior to maturity. Overall, the paper is clearly written and the authors generally are careful and precise in their claims and interpretations of the data. I do, however, have a few concerns with the work, one experimental, one interpretive, and the other about the presentation, that should be addressed.

1. There appears to be a gap in the experiments with respect to the LL genotype. Time series data for vgII3 expression in a number of tissues was done for the EE genotype. This includes multiple immature and mature fish. In addition, for both the EE and LL genotypes, vgII3 expression was measured and compared in immature fish. However, there is no corresponding data on vgII3 expression levels in mature LL fish. As such, the data show that in EE fish, vgII3 expression drops upon maturity and that prior to maturity, vgII3 expression is higher in LL fish. It doesn’t show what happens upon maturity in LL fish. It seems like the authors assume that vgII3 expression also drops upon maturity in LL fish, but this claim is not tested as far as I can tell.

2. The EE and LL genotypes do a good job of explaining variation in time to maturity and the cis-regulatory difference measured between these alleles accounts for much of the expression difference. However, the EE and LL genotypes do a poor job of explaining variation in expression. Given that the proposed model is that differences in genotype affect expression (in cis) and this in turn affects maturity, this lack of explanatory power at the molecular step should be acknowledged. This is not to say that anything is wrong, as finding a molecular association of any kind in such a study is already promising, but I believe it requires discussion as to the possible causes. Some possible causes could be experimental error (i.e. noisy assay), incorrect timing of measurement, or a different molecular step being the key.

3. Framing of the proposed model needs work. In the paper, it is presented in the discussion as the data ‘supporting’ this model. I read this as indicating that the model had previously been establish. However, no reference was given. Alternatively, if this is a new model, phrasing such as the data ‘suggesting’ a model would be more consistent.

In addition, this section contains new analysis and introduces a number of additional molecular steps (e.g. Teads and Yap). This would seem to fit better in the results section of the paper.

Finally, a negative correlation between vgII3 expression and yap1 expression is indicated in the text, but figure S11 doesn’t show this result. Instead S11 shows a negative relationship between vgII3 expression and maturity in a new data set (which is consistent with the rest of the paper and a nice confirmatory step) and a positive relationship between yap1 and maturity. This later relationship seems to be the basis of the new claim by the authors, but it is simply a correlation. While a useful insight that warrants follow-up, it doesn’t rise to the level of an established claim about the molecular relationship between vgII3 and yap1.

Minor

1. While the main text is precise with respect to the experiments, causation is overstated in the abstract. Unless I’m mistaken, inbred lines were not used and individual fish vary at more than just the vgII3 locus. Thus the words mediate and control in the abstract are stronger than what is shown in the paper, which are correlations.

2. The liability model is overstated in the author summary. It is stated as a fact instead of as a hypothesis that arises from the current work.

3. Figure 2A. Muscle and immature ovary are shown but not mentioned in main text. Mature testes is only shown in supplement even though it is focal comparison of the paper. Likewise, stripped testes and ejaculate are shown in the following figure, but the time series data is relegated to the supplement.

Pg. 8. And Figure 3A. Only the difference in vgII3 RNA counts is reported in the text between homozygous genotypes. It would be helpful to have the average values as well for each genotype. In addition, the value reported for heterozygotes in the test (27) doesn’t match the figure.

Figures S8 and S9. To support the claims in the text, p-values showing no difference between EE and LL genotypes are needed. Also, need trend lines on figures to see change with age (or not).

Figure 3B. y-axis label is confusing. From the text, I believe this should be something like ‘% L allele’ of ‘L allele abundance (%)’.

Figure S11. Need SD or SE bars. Or given, that it is only three data points, need to show all of them to get a sense of the expression variation within and between groups.

Reviewer #3: Verta, Primmer, et al present an exceptional study that links cis-regulation of the vestigial-like 3 (vgll3) to alternative life-history traits in Atlantic salmon. They use a series of experiments of seeming monumental effort - 32 familes phenotyped at multiple time points, gene expression measured at 13 time points in 10 tissues – and clever gene expression analyses and visualizations to link expression variation with growth and maturation phenotypes. They show that the early maturation phenotype is associated with downregulation of vgll3 through cis-regulatory variation. The manuscript unfolds nicely with a series of clearly articulated hypotheses and their tests. I think the study and manuscript are exceptional and meet the effort-level and interest/scope typical of high-profile PLoS Genetics papers. I recommend it be published with consideration of the minor points/questions I bring up below.

Minor points:

1. Abstract – I think “variation in” should be included in the first sentence, e.g., “how evolution shapes variation in life histories.” Minor point of preference.

2. I don’t know that the first part of the author summary section is clear enough for all readers to get behind. I think it should be a statement that everyone knows or agrees with. Is “much of the world’s intraspecific biodiversity” “due to alternative strategies to reach reproduction”?

3. Age at maturation as a complex trait – In a previous study cited in wild populations (ref. 4), vgll3 explained ~40% of the variation in sea-age at maturity. It seems like less variation is explained in this controlled setting, surprisingly. I would suggest this be addressed in the discussion.

4. Figure 3B. Shouldn’t the y-axis be “L/E+L allele abundance %”? Not a ratio, but a proportion?

5. Discussion – Excellent discussion, including of cis vs. coding variation as important in life-history trait evolution. It is beyond the scope of this study, but I wonder if it would warrant mention that functional validation, as in manipulation of allele-specific expression, would be an important future direction. To this point, I would not include “functional validation” as a keyword of the study.

**Have all data underlying the figures and results presented in the manuscript been provided?**

Reviewer #1: No: Data underlying figures and results were not provided, but the authors wrote that these data will be available on Dryad and GitHub following manuscript acceptance.

Reviewer #2: No:

Reviewer #3: Yes

PLOS authors have the option to publish the peer review history of their article (what does this mean?). If published, this will include your full peer review and any attached files.

Reviewer #1: Yes: Fabien Duveau

Reviewer #2: No

Reviewer #3: No

---

## [Decision Letter · Decision Letter 1]

12 Aug 2020

Dear Dr Verta,

Thank you very much for submitting your Research Article entitled 'Cis- regulatory differences in isoform expression associate with life history strategy variation in Atlantic salmon' to PLOS Genetics. Your manuscript was fully evaluated at the editorial level and by independent peer reviewers. The reviewers appreciated the attention to an important topic but identified some aspects of the manuscript that should be improved.

We therefore ask you to modify the manuscript according to the review recommendations before we can consider your manuscript for acceptance. Your revisions should address the specific points made by each reviewer.

[LINK]

Yours sincerely,

Juliette de Meaux

Associate Editor

PLOS Genetics

Kirsten Bomblies

Section Editor: Evolution

PLOS Genetics

Dear Authors,

The revised version of the manuscript has now been evaluated by three reviewers. We appreciate the effort deployed to understand the expression changes associating with genetic variation at vgll3. The manuscript is now almost ready for publication. A few modifications are still needed. As indicated by reviewer 1, you need to reformulate some of the conclusions, because your paper associates vgl3 variation with changes in its transcription, but it does not provide a demonstration that this association is causal for the phenotype. It is important this distinction is made clear to the reader. Therefore, I would like to see the suggestions of reviewer 1 be integrated in the final version of the manuscript.Please, also amend accordingly the last sentence of the abstract.

Reviewer's Responses to Questions

**Comments to the Authors:**

Reviewer #1: Please, find my review in the attached document.

Reviewer #2: The authors have addressed all of my initial concerns. The revised model is presented clearly and provides a precise hypothesis for future work. Overall the paper is important, well written, and will make an excellent contribution to the field.

Reviewer #3: The study by Verta et al provides an exceptional example of genotype-phenotype association for a critical life-history trait, age at maturity in Atlantic salmon, a species of great economic interest. The authors thoroughly revised their study according to reviewers and editor suggestions and performed new experiments that changed their results and model, but did not change their conclusions and the significance of their study. The authors sufficiently addressed my questions, comments, and concerns in their revised manuscript. They revised the mechanism linking age at maturity to isoform/splice variation of vgll3 rather than allele specific expression adding new data and a revised model. I think this revised article would make an important contribution of interest to the broad readership of PLoS Genetics.

**Have all data underlying the figures and results presented in the manuscript been provided?**

Reviewer #1: **No: **The authors declared that the data will be available following manuscript acceptance.

Reviewer #2: Yes

Reviewer #3: Yes

PLOS authors have the option to publish the peer review history of their article (what does this mean?). If published, this will include your full peer review and any attached files.

Reviewer #1: **Yes: **Fabien Duveau

Reviewer #2: No

Reviewer #3: No

---

## [Editor Report · Decision Letter 2]

17 Aug 2020

Dear Dr Verta,

We thank you for adding these last minor revisions and are pleased to inform you that your manuscript entitled "Cis- regulatory differences in isoform expression associate with life history strategy variation in Atlantic salmon" has been editorially accepted for publication in PLOS Genetics. Congratulations!

Yours sincerely,

Juliette de Meaux

Associate Editor

PLOS Genetics

Kirsten Bomblies

Section Editor: Evolution

PLOS Genetics

Comments from the reviewers (if applicable):

**Data Deposition**

http://datadryad.org/submit?journalID=pgenetics&manu=PGENETICS-D-19-01983R2

**Press Queries**

---

## [Editor Report · Acceptance letter]

24 Sep 2020

PGENETICS-D-19-01983R2 

Cis- regulatory differences in isoform expression associate with life history strategy variation in Atlantic salmon 

Dear Dr Verta, 

We are pleased to inform you that your manuscript entitled "Cis- regulatory differences in isoform expression associate with life history strategy variation in Atlantic salmon" has been formally accepted for publication in PLOS Genetics! Your manuscript is now with our production department and you will be notified of the publication date in due course.

With kind regards,

Jason Norris

PLOS Genetics

On behalf of:
